# Intraspecific Variability in Proteomic Profiles and Biological Activities of the Honey Bee Hemolymph

**DOI:** 10.3390/insects14040365

**Published:** 2023-04-06

**Authors:** Salma A. Elfar, Iman M. Bahgat, Mohamed A. Shebl, Mathieu Lihoreau, Mohamed M. Tawfik

**Affiliations:** 1Zoology Department, Faculty of Science, Port Said University, Port Said 42526, Egypt; 2Department of Plant Protection, Faculty of Agriculture, Suez Canal University, Ismailia 41522, Egypt; 3Research Center on Animal Cognition, Center for Integrative Biology, Centre National de la Recherche Scientifique, University Paul Sabatier, 31062 Toulouse, France

**Keywords:** honeybee, proteome, anticancer, antimicrobial, antioxidant

## Abstract

**Simple Summary:**

Insect hemolymph is equivalent to blood in higher vertebrates. It is the main site for immune responses, mediates nutrient transportation to organs and tissues, and has antimicrobial and antioxidant properties. Hemolymph can thus provide information about the health status of an insect. Here we report intraspecific variation in hemolymph properties of Western honey bees *Apis mellifera* sampled in four locations providing different diets across Egypt. Bees that had access to a rich and varied diet had higher protein concentrations and levels of biological activities in their hemolymph than bees that were only fed sucrose solution. This suggests hemolymph analyses could be used as a powerful indicator for monitoring bee populations, with the aim of improving their health and pollination efficiency.

**Abstract:**

Pollinator declines have raised major concerns for the maintenance of biodiversity and food security, calling for a better understanding of environmental factors that affect their health. Here we used hemolymph analysis to monitor the health status of Western honey bees *Apis mellifera*. We evaluated the intraspecific proteomic variations and key biological activities of the hemolymph of bees collected from four Egyptian localities characterized by different food diversities and abundances. Overall, the lowest protein concentrations and the weakest biological activities (cytotoxicity, antimicrobial and antioxidant properties) were recorded in the hemolymph of bees artificially fed sucrose solution and no pollen. By contrast, the highest protein concentrations and biological activities were recorded in bees that had the opportunity to feed on various natural resources. While future studies should expand comparisons to honey bee populations exposed to more different diets and localities, our results suggest hemolymph samples can be used as reliable indicators of bee nutrition.

## 1. Introduction

Bees are key pollinators [1,2] whose widespread declines have raised major concerns for the maintenance of terrestrial biodiversity and food security [3]. Over the past decades, their accelerated losses have raised the need for a better understanding of the environmental stressors that affect population growth and their mechanisms of action, for instance, through large-scale monitoring of bee colony health statuses across different habitats [4].

Like all insects, bees have an open circulatory system composed of hemolymph, which is equivalent to blood in higher vertebrates [5]. Hemolymph is vital to bees because it mediates the distribution of nutrients through the body by supplying tissues and organs [6]. It is the main site for defense against infections [7] and possesses antimicrobial, antioxidant and anticancer activities [8]. The principal components of hemolymph are water, carbohydrates, proteins, inorganic salts, lipids, hormones and immune cells. Variations in protein contents, which are strongly linked to nutrition [9,10,11], can inform about the physiological and immune status of bees [5,12]. For instance, high hemolymph protein levels minimize the susceptibility of honey bees to pathogens [13,14]. By contrast, low hemolymph protein levels are a signature of poor health status. Therefore, hemolymph analysis can serve as a practical and powerful means for monitoring bee health status. Ultimately, a better understanding of hemolymph variations across individuals and populations may also help better assess its value as a potential therapeutic compound, as recently suggested (i.e., anticancer and hemolytic activities) [15].

Previous research has shown that the hemolymph of the honey bee *Apis mellifera* varies in protein composition across individuals of the same population, especially within castes and among different developmental stages [16]. However, little is known about the variation in hemolymph protein composition among workers with consideration of diet variations. Moreover, these studies have often used different solvents to extract hemolymph samples, which makes it difficult to compare since the nature of solvents affects the solubility and structure of proteins and, thus, the content and biological activities of hemolymph samples [17].

Here we evaluated intraspecific variation in proteomic contents and biological activities of honey bees (*A. mellifera*) hemolymph collected from four localities across Egypt, characterized by contrasted diets (natural vs. artificial diets). We extracted hemolymph samples from 400 honey bees (100 per location) using two different solvents and quantified their proteomic content and potential anticancer, antibacterial, antioxidant, and hemolytic activities.

## 2. Materials and Methods

### 2.1. Chemicals

Chemicals were purchased from Sigma Chemical Co. (St. Louis, MO, USA). Hydrophilic Phosphate buffer saline (PBS) and hydrophobic Dimethyl sulfoxide (DMSO) were used as solvents for the extraction of honey bee hemolymph samples. These two complementary solvents were used in order to extract as many molecules as possible.

### 2.2. Sample Collection and Preparation

We exclusively worked on forager honey bees (hybrids of the subspecies *A. mellifera lamarckii, A. mellifera carnica* and *A. mellifera ligustica*). Sweep nets were used to collect bees of unknown age in the different study sites (ca. 100 bees per location) from May to July 2021. Foragers were caught while flying around food sources. All the foragers from each site were collected on the same day. Bees from the faculty of science of Port Said Governorate originated from colonies enclosed in a big flight tent and maintained in an artificial beehive with access to sugar syrup only for the duration of the experiment (from the beginning of May to the end of July). Bees from Ismailia, Suez governorates and Saint Catherine had access to different cultivated plants (see details in Table 1), thus providing diverse nutritional resources. The bees were cold-anesthetized (kept at −20 °C) for 5 min to facilitate hemolymph extraction in live individuals [18].

### 2.3. Hemolymph Collection and Preparation

Hemolymph was extracted according to Łoś and Strachecka [19] and Basseri [20]. Here, we sampled near the coxal membrane using sterile insulin syringes and pressing the abdomen. Clear and slightly yellow hemolymph was drawn out from the wound. If cloudy yellow intestinal contents were taken, the sample was discarded. The hemolymph samples were kept in sterile Eppendorf tubes and stored at −20 °C until lyophilization. About 9–14 µL of hemolymph were extracted per bee. All the samples from a given site were then pooled, yielding about 1000–1200 µL of hemolymph. After lyophilization, this resulted in about 100 mg per location. Equal weights of lyophilized hemolymph were dissolved in DMSO or PBS.

### 2.4. Protein Estimation

Protein concentration was measured in mg/mL using a Thermo Scientific™ (Waltham, MA, USA) Nano Drop™ One Micro volume UV-Vis Spectrophotometer. Bovine serum albumin (BSA) was added as a standard in each sample (2 µL, 1 mg/mL) [21]. Three samples were analyzed per site.

### 2.5. SDS-Polyacrylamide Gel Electrophoresis

Protein gel profiles were obtained to investigate variations in the types of proteins among samples by using sodium dodecyl sulfate-polyacrylamide gel electrophoresis (SDS-PAGE) for separating proteins depending on their molecular weights according to Laemmli [22]. SDS-PAGE was performed at a total acrylamide concentration of 12%. The samples were mixed with the solubilizing buffer that contains 62.5 mM Tris-HCl (pH 6.8), 20% glycerol, 25 (*w*/*v*) SDS, 0.5% 2-mercaptoethanol and 0.01% bromophenol blue, then heated for 4 min at 95 °C. The samples were then immediately loaded into wells (15 µL of the sample with a protein concentration of 100 µg/mL per well). Electrophoresis was carried out at a constant 35 mA for 2 h using Consort N.V. (Turnhout, Belgium) mini vertical system with a running buffer. The gel was stained with 0.1% Coomassie Brilliant Blue (R-250) for the protein bands to be visualized.

### 2.6. HPLC (High-Performance Liquid Chromatography) Analysis

Reversed-phase HPLC chromatograms of hemolymph extracts were used to separate the protein peaks based on their retention time. Each of the lyophilized hemolymph samples was dissolved in PBS or DMSO at the same concentration (5 mg/mL). 70 µL samples from each location were analyzed using YL9100 HPLC System under the following conditions: C18 column (Promosil C18 Column 5 μm, 150 mm × 4.6 mm) as stationary phase and acetonitrile (ACN) gradients in the range of 10% to 100% acetonitrile in water for 50 min as mobile phase at flow rate 1 mL/min and by using UV detector at wavelength 280 nm [20].

### 2.7. Anticancer Activity (MTT-Assay)

The anticancer activities of the hemolymph extracted in PBS and DMSO were measured in vitro. Human hepatocellular carcinoma (HepG2) and human cervical carcinoma (HeLa) cells were obtained from the Holding company for biological products and vaccines (VACSERA), Giza, Egypt. Cells were seeded in 96-well plate for 24 h (5 × 10^3^ cells/well). After incubation, the cells were treated with serial concentrations of the 100 μL hemolymph extracts in each solvent (31.25, 62.5, 125, 250, 500 and 1000 µg/mL) and incubated at 37 °C in a 5% CO_2_ atmosphere for 48 h.

The cytotoxic effect of extracts on the proliferation of cells was evaluated using the 3-[4,5-methylthiazol-2-yl]-2,5-diphenyl-tetrazolium bromide (MTT) assay [23]. After 48 h of treatment, the medium, including MTT dye, was added to cells and incubated at 37 °C for 4 h to enable the production of formazan crystals in the viable cells only. DMSO was used to solubilize the formazan crystals. The absorbance was then measured at 540 nm using a Bio-Tek microplate reader ELISA. Each experiment was performed in triplicate, and the percentage of cell viability was calculated as follows:Cell viability (%) = (**A_T_**/**A_C_**) × 100
where **A_T_** is the absorbance of treated cells, and **A_C_** is the absorbance of the control cells (untreated cells).

The concentration that inhibits the growth of 50% of the cells (IC_50_) values was determined for each sample from a dose-response curve between dose concentration (*x*-axis) and cell inhibition percentage (*y*-axis).

### 2.8. Antimicrobial Activity Screening

The antimicrobial activity of hemolymph extracts was assessed using agar well diffusion [24]. Antimicrobial activity was determined against two Gram-positive bacteria (*Bacillus subtilis* and *Staphylococcus aureus*) and two Gram-negative bacteria (*Escherichia coli* and *Salmonella typhimurium*). Briefly, the agar plate surface was inoculated by spreading a volume of the microbial inoculum over the entire agar surface, then making a hole with a diameter of 6 to 8 mm aseptically using a sterile tip. A volume of about 20–100 µL of each extract (10 mg/mL) or control at a known concentration was introduced into the well. The agar plates were then incubated at suitable conditions depending on the microorganism used for the test. Gentamycin was used as a positive control, while PBS and DMSO were used as negative controls. The antibacterial activities of the extracts were expressed as inhibition zones in millimeters [25,26].

### 2.9. DPPH Radical Scavenging Assay

The antioxidant properties of hemolymph were assessed in vitro. The radical scavenging capacity of the hemolymph extracts on the stable free radical 1,1-Diphenyl-2-picrylhydrazyl (DPPH) was tested according to Braca [27]. Briefly, 100 μL of DPPH methanolic solution (0.004% in methanol) were mixed with 0.3 mL of sample extracts (1 mg/mL) or standard and incubated in the dark for 30–60 min at 25 °C. Ascorbic acid (Vitamin C) was used as a standard (positive control), and the absorbance was measured at 515 nm by using a spectrophotometer. A negative control was prepared by adding 2.7 mL of DPPH in 0.3 mL of the solvent used in the extract (AC). The antioxidant activity of the samples was calculated as follows [28,29]:Antioxidant activity (%) = [(**A_C_** − **A_E_**)/**A_C_**] × 100
where **A_C_** is the mean absorbance of negative control, and **A_E_** is the mean absorbance of the extract or standard.

### 2.10. Hemolytic Activity Assay

We assessed the hemolytic activity of bee hemolymph against human red blood cells to elucidate its selectivity and safety as a potential therapeutic agent. We compared the ability of extracts of 100 μL at concentrations 125, 250, 500, 1000, 2000 and 4000 µg/mL to lyse human erythrocytes [30]. Fresh blood samples were collected in test tubes containing anticoagulant (EDTA) and then centrifuged for 5 min at 2000 rpm. Samples were washed several times with sterile PBS. Various concentrations of the extracts were added to RBCs and incubated for an hour at room temperature. The samples were then centrifuged for 5 min at 10,000 rpm, and the absorbance of the released hemoglobin was measured at 570 nm. 10% Triton X-100 was used as a positive control (100% hemolysis), and sterile PBS and DMSO as negative controls (0% hemolysis). This experiment was carried out three times, and the hemolysis percentage was calculated for each sample by using the equation:Hemolysis (%) = 100 × [(**A_S_** − **A_N.C_**)/(**A_P.C_** − **A_N.C_**]
where **A_S_** is the absorbance of the sample, **A_N.C_** is the absorbance of the negative control, and **A_P.C_** is the absorbance of the positive control.

### 2.11. Statistical Analysis

All analyses were done in SPSS 22.0. Pairwise comparisons between parameters of hemolymphs extracted in PBS and DMSO were made with Student’s *t*-tests. Multiple comparisons between parameters of hemolymphs from bees of the four study sites were made with One-way ANOVAs followed by a Tukey’s test. When the *p*-value was lower than 0.05, the difference in the means of the samples was considered statistically significant. Means are reported with standard errors (mean ± SE).

## 3. Results

### 3.1. Naturally Fed Honey Bees Had Higher Protein Concentrations in Hemolymph

The amount of proteins found in each hemolymph extract was measured using spectrophotometry (Figure 1). Among the DMSO dissolved samples, extracts from site **B** had higher protein concentrations (0.12 ± 0.0006 mg/mL, *n* = 3 samples) than extracts from all the other sites. Among the PBS dissolved samples, extracts from site **D** had the highest protein concentrations (0.19 ± 0.03mg/mL, *n* = 3 samples). Extracts from site **A** had a significantly lower protein concentration than extracts from all the other sites, being dissolved in PBS (0.006 ± 0.0005 mg/mL, *n* = 3 samples) or DMSO (0.02 ± 0.0003 mg/mL, *n* = 3 samples). Protein concentrations thus varied among the hemolymph extracts of bees fed different diets. The lowest concentrations were consistently recorded for the artificially fed bees in site **A**.

### 3.2. Honey Bee Hemolymph Showed Different Protein Composition across Sites

Quantitative analysis of proteins in the hemolymph extracts revealed a variety of protein bands ranging in mass from 5 to ~250 kDa (Figure 2).

The SDS profile of the samples dissolved in PBS (Figure 2A) revealed six bands with molecular weights ~40, ~50, ~60, ~100, ~180 and ~250 kDa common to extracts from all sites. A band with a molecular weight of ~10 kDa was only recorded in extracts from sites A and B. A band with a molecular weight of ~17 kDa was only recorded in extracts from sites C and D. Three bands with molecular weights ~5, ~30, and ~70 kDa were specific to extracts from site D.

Among the samples dissolved in DMSO (Figure 2B), bands with molecular weights ranging from ~17 to ~250 kDa were observed in all sites. A band with a molecular weight of ~5 kDa was only observed in extracts from site D.

### 3.3. HPLC Analyses Were Then Used to Separate the Proteins of the Hemolymph Extracts According to Their Peaks Expressed in Retention Time

RP-HPLC chromatograms are supplemented in the Appendix A. The chromatograms of the hemolymph extracts dissolved in PBS revealed five peaks that were common to extracts from sites B, C and D at retention times of 7.6, 12.4, 19.9, 25.6 and 43.0 min. The chromatograms of extracts dissolved in DMSO revealed another five peaks common to all sites at retention times of 22.1, 28.7, 30.4, 33.2 and 38.7 min (Table 2).

### 3.4. Honey Bee Hemolymph Suppresses the Growth of HepG2 and HeLa Cells

The cytotoxic effects of hemolymph were evaluated against the viability of HepG2 and HeLa. The viability of the two cancer cell lines was inhibited in a dose-dependent manner after being treated with different concentrations of the hemolymph extracts in either PBS or DMSO after 48 h of treatment (Figure 3). Overall, DMSO-dissolved extracts possessed higher anti-proliferative activity than PBS-dissolved extracts (Figure 4).

Regarding extracts dissolved in DMSO, hemolymph from site C showed the highest cytotoxic effects against HepG2 and HeLa cells (IC_50_ = 52.03 & 97.95 µg/mL, respectively). By contrast, the lowest cytotoxic effect was recorded for hemolymph from site A (IC_50_ = 463.36 & 454.02 µg/mL) (Figure 4).

When considering extracts dissolved in PBS, the highest cytotoxic effect was recorded for hemolymph from site D (IC_50_ = 164.5 & 169.7 µg/mL against HepG2 and HeLa, respectively) and the lowest cytotoxic effect was observed in hemolymph from sites A and B (IC_50_ = 649.36 & 533.08 µg/mL against HepG2 and HeLa respectively) (Figure 4).

### 3.5. Honey Bee Hemolymph Inhibits the Growth of Gram-Positive and Gram-Negative Bacteria

The antimicrobial activity of the hemolymph extracts from sites A, B, C and D was analyzed against two Gram-positive bacteria and two Gram-negative bacteria using the well diffusion method (Table 3). Overall, PBS samples showed higher antibacterial activity than DMSO samples (*p* < 0.05). For both solvents (Table 3), the highest antibacterial activities were observed for the extracts from site B against *E. coli* with inhibition zones 40 mm and 38 mm, respectively. However, extracts from site A showed the lowest antibacterial activities against all tested bacteria, with an inhibition zone ranging from 19 to 28 mm (Figure 5).

### 3.6. Honey Bee Hemolymph Scavenges DPPH Free Radical

The antioxidative activity of the hemolymph extracts on DPPH free radicals showed a potential radical scavenging activity in both PBS and DMSO solvents. Overall, PBS-dissolved extracts displayed higher antioxidant properties than DMSO-dissolved extracts (Figure 6) (*p* < 0.05).

Considering PBS- and DMSO-dissolved extracts independently, there were significant differences between sites (*p* < 0.001). Extracts from site D consistently possessed the highest antioxidant activity in both solvents (*p* < 0.001), while extracts from site A possessed the lowest scavenging activity (Figure 6).

### 3.7. Honey Bees Possess Low Hemolytic Activity against Human Erythrocytes

Hemolytic activities of the honey bee hemolymph extracts at different concentrations were evaluated on human erythrocytes. None of the extracts displayed hemolytic activity against erythrocytes when compared to negative and positive controls (Figure 7).

## 4. Discussion

Our study describes intraspecific variations in the protein profiles, concentrations and biological activities of the hemolymph of honey bees (*A. mellifera*) collected from different localities in Egypt chosen to offer contrasted diets. The highest protein concentrations, diversity and bioactivity levels were recorded in the hemolymph of bees that had the opportunity to feed on various natural resources when compared to bees fed sucrose solution and no pollen.

The fact that high protein concentrations were recorded in the hemolymph of bees fed natural resources is consistent with the observation of a negative impact of a sugar solution diet on the protein concentration of honey bee hemolymph [31]. Other previous studies have reported the influence of different diets of honey bees on their hemolymph structure [32,33], indicating that the protein content of hemolymph can be used to assess the efficacy of protein diets and pollen quality [10,34,35,36]. Our study further shows hemolymph proteomic profile can also be used as an indicator of bee health since high protein concentrations extend bee life span and improve immunity and resistance to pathogens [37,38,39].

Our analysis also revealed important variations in protein concentration depending on the solvent used for extraction. PBS and DMSO were chosen for their complementarity: PBS extracts water-soluble proteins (hydrophilic) [40], while DMSO extracts lipid-soluble proteins (hydrophobic) [41]. Proteins structure, stability and solubility are markedly affected by the polarity of solvents [17], which may explain why hemolymph biological activities greatly varied between PBS and DMSO dissolved samples. We, therefore, recommend future detailed analyses of hemolymph protein contents should use these two complementary solvents for comparative purposes.

The SDS-PAGE gel of the hemolymph of our extracts also showed variations in their protein bands. In general, more protein bands were recorded using DMSO as compared to PBS. Bands with molecular weight ~17 kDa were detected in most extracts dissolved in PBS and DMSO. These bands are in the same range as a class of lectins, which is a common group of proteins in arthropods [42]. Insect lectins have several important roles in immune response, homeostasis and binding for the storage or transport of carbohydrates [43,44]. Insect lectins may also possess antimicrobial, antioxidant and anticancer activities and can be considered as a potential natural bioactive agent [45,46,47,48,49]. Protein bands with molecular weight ~60 kDa were also found in all the tested PBS- and DMSO-dissolved extracts. Similar antimicrobial proteins have been fractionated from cockroach hemolymph [20].

A hemocyanin-like protein band ~70 kDa was also detected in all DMSO dissolved extracts but also some of those dissolved in PBS. Hemocyanin is an oxygen transport protein that has a wide range of biological activities, including antioxidant, antiparasitic, antimicrobial and anticancer activities [48,50,51]. Protein bands with molecular weight ~110 kDa were found in all PBS- and DMSO-dissolved extracts. These bands evoke hexamerins, which are storage proteins that are derived from hemocyanin [52]. The protein bands with molecular weight ~180 kDa were detected in all PBS- and DMSO-dissolved extracts. These bands are in the same range of vitellogenin in honey bee hemolymph [16]. Vitellogenin is a large female-specific gluco-lipoprotein that can act as female nutrient storage and also lipid carrier protein [53,54]. Interestingly, honey bee vitellogenin was shown to have antimicrobial, antioxidant and immunological activities [55].

Protein bands with molecular weight > 250 kDa have also been observed in the proteomic profiles of all PBS and DMSO hemolymph extracts. These bands likely correspond to apolipoprotein, a lipid transporter lipoprotein [56]. These findings agree with Ba et al. (1987) [57] and Chan et al. (2016) [16], who revealed similar bands in honey bee hemolymph.

Interestingly, chromatograms of honey bee hemolymph collected from Saint Catherine showed the highest peak areas among all sites, which is an indicator of the amount of protein. This reinforces the results of the SDS-PAGE gel, which reveals that the protein bands of extract from this location had the highest protein concentrations based on the color intensity of the bands. Saint Catherine is the most floristically diverse region of the Middle East. It is a naturally protected area that accounts for nearly one-fourth of the total flora of Egypt. Such vegetation diversity lets bees in this region have a foraging advantage which may probably enrich the protein content of honey bee hemolymph [58,59,60,61].

Both PBS and DMSO dissolved extracts revealed a relative degree of cytotoxicity against HepG2 and HeLa in a dose-dependent manner. There was variation in the cytotoxicity between the hemolymph samples in the two solvents against the two tested cell lines. This variation may be related to the significant differences in their protein profiles and antioxidant activities. The higher cytotoxicity of DMSO-dissolved extracts over PBS-dissolved extracts against both cell lines may be attributed to the role of DMSO in dissolving lipoproteins [41].

The diversity and prevalence of the bioactive proteins such as lectins, hemocyanin, vitellogenin, hexamerins and apolipophorin in PBS and DMSO extracts may elucidate variations in the biological activities of hemolymph described in our results. Hemolymph showed prominent antimicrobial activities against different species of Gram-positive and Gram-negative bacteria. Honey bees, like other invertebrates, possess a very effective innate immune mechanism to defend themselves against microorganisms and pathogens [62], and deficiencies in hemolymph proteins affect their ability to resist diseases [63]. Consequently, feeding on natural resources likely has a great impact on hemolymph bioactivities [64,65].

None of the hemolymph extracts, be they dissolved in PBS or DMSO, showed hemolytic activity when experimented against human erythrocytes at different tested concentrations. This observation agrees with Mokarramat-Yazdi et al. (2021) [15], who reported an absence of hemolytic activity in honey bee hemolymph. Interestingly, in this study, the authors suggested a potential anticancer effect of hemolymph mixed with herbals in mice models. In this regard, honey bee hemolymph is a potential natural selective therapeutic agent in cancer treatment.

## 5. Conclusions

Our comparative analysis reveals significant intraspecific variability in the composition and concentration of hemolymph of honey bees sampled in Egypt. The hemolymph of the bees that could feed on cultivated plants and wild medicinal plants possessed relatively higher protein concentrations, as well as higher antimicrobial, antioxidant and anticancer properties than that of bees reared only on sugar solutions in artificial conditions. Future studies should expand our approach to more localities and diets in order to clarify the link between diet and hemolymph properties, as well as explore the potential therapeutic usage of honey bee hemolymph.

## Figures and Tables

**Figure 1 insects-14-00365-f001:**
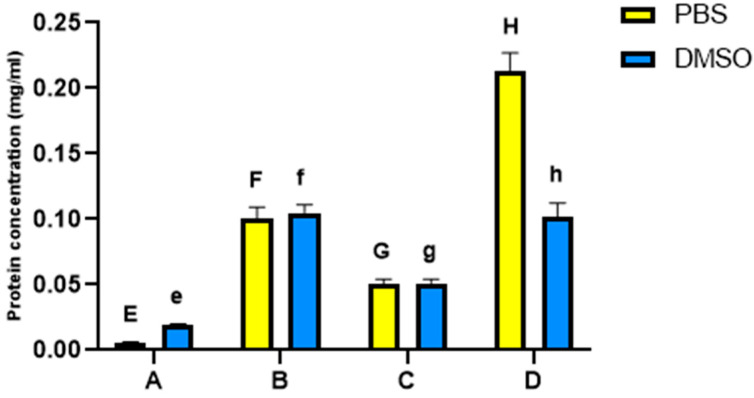
Protein concentrations in mg/mL for honey bee hemolymph extracted in PBS or DMSO where (**A**): bees from Port Said, (**B**): bees from Ismailia governorate, (**C**): bees from Suez governorate, (**D**): bees from Saint Catherine. Bars and error bars represent the mean values ±SE obtained from triplicate measurements. Different letters above bars indicate significant differences (Tukey’s test, *p* < 0.05): uppercase letters show comparisons across PBS dissolved samples; lowercase letters show comparisons across DMSO dissolved samples.

**Figure 2 insects-14-00365-f002:**
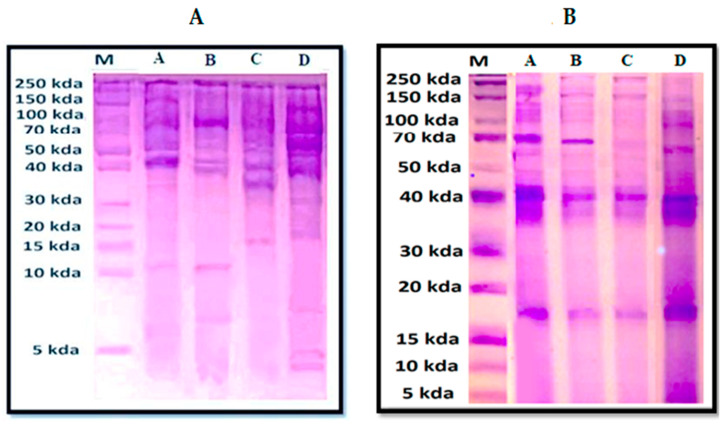
Photos of SDS-PAGE gels for honey bee hemolymph extracts dissolved in PBS (**A**) or DMSO (**B**) where (**A**): bees from Port Said, (**B**): bees from Ismailia governorate, (**C**): bees from Suez governorate, (**D**): bees from Saint Catherine. **M**: marker. The molecular weights of the proteins and their amounts are expressed by the color intensity of the protein bands.

**Figure 3 insects-14-00365-f003:**
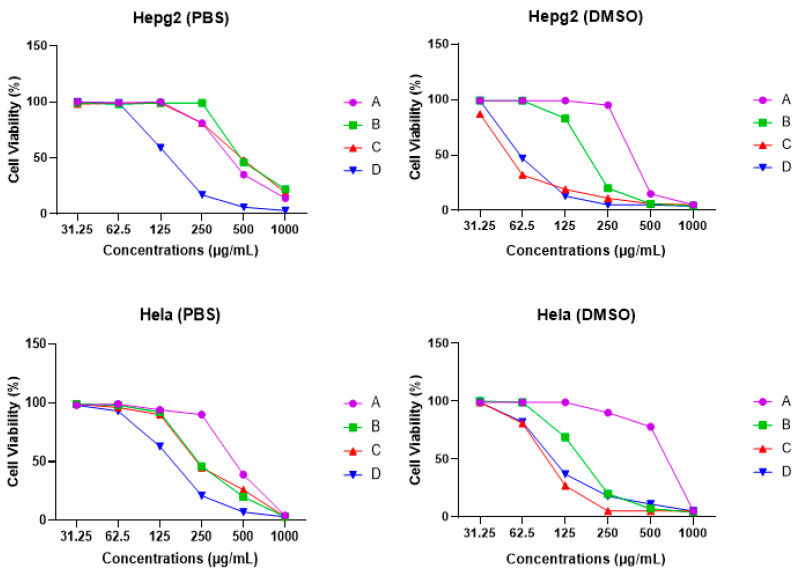
Effects of honey bee hemolymph extracts dissolved at different concentrations in PBS or DMSO on human cancer cell proliferation (HepG2 and HeLa), where (**A**): bees from Port Said, (**B**): bees from Ismailia governorate, (**C**): bees from Suez governorate, (**D**): bees from Saint Catherine.

**Figure 4 insects-14-00365-f004:**
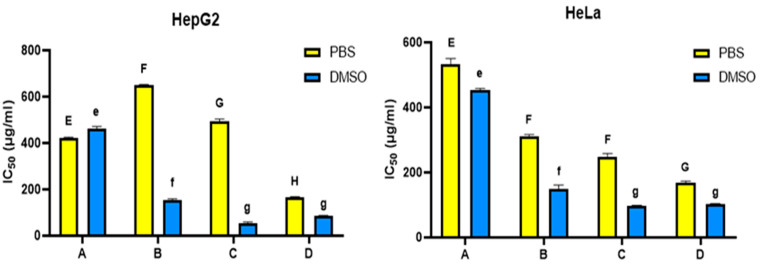
IC_50_ in microgram/mL of hemolymph extracts of honey bees dissolved in PBS or DMSO against Hepg2 and HeLa cell lines, where (**A**): bees from Port Said, (**B**): bees from Ismailia governorate, (**C**): bees from Suez governorate, (**D**): bees from Saint Catherine. Bars and error bars represent the mean values ± SE obtained from triplicate measurements. Different letters above bars indicate significant differences (Tukey’s test, *p* < 0.05): uppercase letters show comparisons across PBS dissolved samples; lowercase letters show comparisons across DMSO dissolved samples.

**Figure 5 insects-14-00365-f005:**
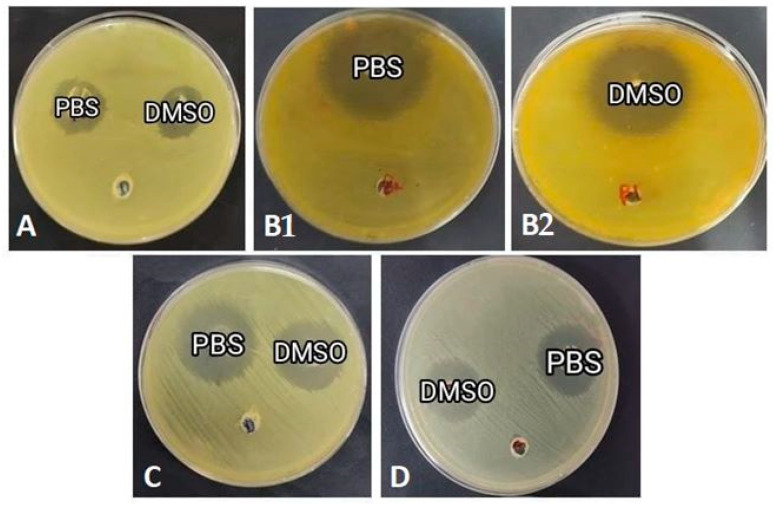
Antimicrobial activity of honey bee hemolymph extracted in PBS or DMSO against *E. coli*, where (**A**): bees from Port Said, (**B1**,**B2**): bees from Ismailia governorate, (**C**): bees from Suez governorate, (**D**): bees from Saint Catherine.

**Figure 6 insects-14-00365-f006:**
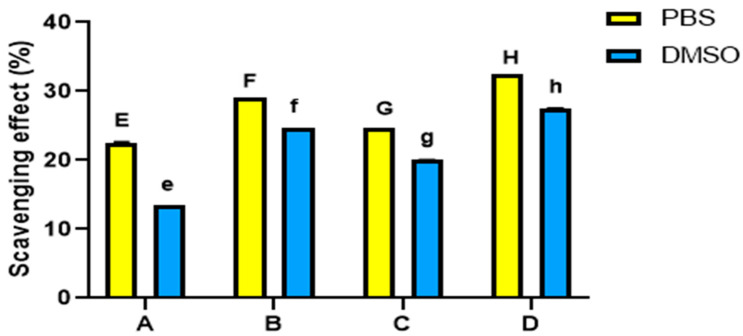
DPPH radical scavenging activity of honey bee hemolymph extracted in PBS or DMSO where (**A**): Bees from Port Said, (**B**): bees from Ismailia governorate, (**C**): bees from Suez governorate, (**D**): bees from Saint Catherine. Bars and error bars represent the mean values ± SEM obtained from triplicate measurements. Different letters above bars indicate significant differences (Tukey’s test, *p* < 0.05): uppercase letters show comparisons across PBS dissolved samples; lowercase letters show comparisons across DMSO dissolved samples.

**Figure 7 insects-14-00365-f007:**
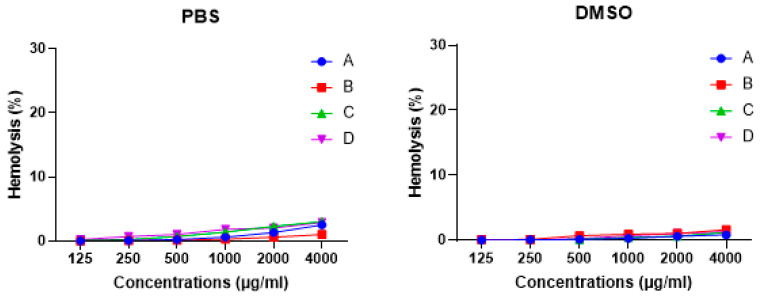
Hemolytic activity of honey bee hemolymph extracted in PBS or DMSO at different concentrations against erythrocytes where (**A**): bees from Port Said, (**B**): bees from Ismailia governorate, (**C**): bees from Suez governorate, (**D**): bees from Saint Catherine.

**Table 1 insects-14-00365-t001:** Localities, feeding diet and types of plantations in the study areas.

Symbol	Locality	Main Feeding Diet	Main Plantation
A	Port said Governorate 31.259 N 32.27 E	Artificial (Sucrose syrup)	N.A
B	Ismailia Governorate 30.61 N 32.25 E	Natural	Cotton
C	Suez Governorate 30.02 N 32.34 E	Natural	Corn, cucumber, okra and sesame
D	Saint Catherine 28.56 N 33.94 E	Natural	Medicinal plants (thyme and sial tree)

**Table 2 insects-14-00365-t002:** RP-HPLC chromatogram peaks profiles of honey bee hemolymph extracts obtained in PBS or DMSO.

Retention Time (min)			Peak Area (×10^2^ mV.s)					
	A		B		C		D	
	PBS	DMSO	PBS	DMSO	PBS	DMSO	PBS	DMSO
2.9	N.A	N.A	1.8	N.A	12.3	N.A	N.A	N.A
4.0	N.A	N.A	N.A	27.7	N.A	1167	N.A	0.2
4.3	N.A	N.A	N.A	N.A	14.4	N.A	3386	N.A
5.0	N.A	N.A	N.A	838.7	N.A	N.A	N.A	397
6.7	36.4	N.A	3.9	N.A	1.0	N.A	N.A	N.A
7.6	N.A	N.A	4.6	N.A	0.8	N.A	163.5	N.A
10.3	N.A	N.A	N.A	N.A	4.8	N.A	247.5	N.A
11.3	N.A	N.A	2.3	N.A	3.9	N.A	N.A	N.A
12.4	N.A	N.A	10.9	N.A	2.5	N.A	1265	N.A
14.4	N.A	N.A	5.9	N.A	N.A	N.A	105.3	N.A
19.9	N.A	N.A	4.1	N.A	5.0	N.A	363.7	N.A
22.1	N.A	0.3	N.A	1.5	N.A	0.6	N.A	2.6
23.2	N.A	N.A	N.A	3.6	N.A	1.0	N.A	N.A
23.5	N.A	0.9	N.A	N.A	N.A	N.A	N.A	1.2
25.6	N.A	N.A	9.1	N.A	18.0	N.A	770.9	N.A
28.7	N.A	0.6	N.A	2.6	N.A	0.3	N.A	1.8
28.8	43.2	N.A	2.3	N.A	4.7	N.A	N.A	N.A
30.3	94.8	N.A	1.8	N.A	5.2	N.A	N.A	N.A
30.4	N.A	1.3	N.A	0.5	N.A	0.2	N.A	0.4
33.2	N.A	0.2	N.A	0.3	N.A	0.3	N.A	0.2
34.8	N.A	N.A	N.A	0.8	N.A	N.A	N.A	1.4
36.5	N.A	1.0	N.A	0.7	N.A	N.A	N.A	0.2
38.1	N.A	N.A	9.1	N.A	3.0	N.A	N.A	N.A
38.7	N.A	10.5	N.A	0.2	N.A	0.1	N.A	0.1
40.7	N.A	N.A	N.A	0.1	N.A	N.A	N.A	0.1
43.0	N.A	N.A	4.4	N.A	2.6	N.A	99.1	N.A
47.7	N.A	N.A	14.6	N.A	10.7	N.A	N.A	N.A

Where, (**A**): bees from Port Said, (**B**): bees from Ismailia governorate, (**C**): bees from Suez governorate, (**D**): bees from Saint Catherine, (**N.A**: not applicable).

**Table 3 insects-14-00365-t003:** Inhibition zone (mm) of honey bee hemolymph extracted in PBS or DMSO on various types of bacteria (*Bacillus subtilis*, *Staphylococcus aureus*, *Escherichia coli* and *Salmonella typhimurium*).

	Sample	PBS				DMSO				CON.
Pathogen		A	B	C	D	A	B	C	D	
*Bacillus subtilis*		28 ± 1.2 ^a^	34 ± 1.0 ^b^	30 ± 0.9 ^a^	27 ± 0.9 ^a^	26 ± 0.6 ^a^	32 ± 1.6 ^b^	31 ± 1.5 ^ab^	25 ± 1.0 ^ac^	22 ± 0.6
*Staphylococcus aureus*		23 ± 1.0	25 ± 2.0	28 ± 0.9	28 ± 1.0	21 ± 1.6	24 ± 0.9	23 ± 1.0	25 ± 1.9	15 ± 0.9
*Escherichia coli*		21 ± 1.3 ^a^	40 ± 1.5 ^b^	32 ± 0.5 ^c^	25 ± 1.3 ^a^	19 ± 1.3 ^a^	38 ± 1.0 ^b^	27 ± 1.7 ^c^	23 ± 0.7 ^ac^	17 ± 1.0
*Salmonella typhimurium*		22 ± 1.3	N.A	26 ± 1.3	N.A	20 ± 1.5	N.A	25 ± 2.2	N.A	23 ± 0.7

Where **NA** = no activity, (**A**): bees from Port Said, (**B**): bees from Ismailia governorate, (**C**): bees from Suez governorate, (**D**): bees from Saint Catherine, **CON**: Gentamycin as a positive control. Mean values ± SE were obtained from triplicate measurements. Values with different superscript letters (a–c) in the same raw and for a given solvent (PBS or DMSO) are significantly different across locations (Tukey test, *p* < 0.05).

## Data Availability

Data available on request due to restrictions, e.g., privacy or ethics.

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
