# Peer review of "Intraspecific Variability in Proteomic Profiles and Biological Activities of the Honey Bee Hemolymph"

_insects, 2023, doi:10.3390/insects14040365_

Round 1

Reviewer 1 Report

ABSTRACT:

Abstract is weak because it does not reflect what was done and obtained in the study. For example, authors only mentioned „biological activities“ without explanation of multiple strong analyses. I also do not agree that the analyses of hemolymph done in this study present „a cheap, simple, yet powerful approach“ (they are not cheap and simple). Abstract has to be improved by incorporating exact details of the study. Moreover, I am wondering why the only conclusion is: that hemolymph samples can be used as reliable indicators of bee nutrition and health.

INTRODUCTION:

- Line 64: Instead of „some biological activities“, I suggest authors to itemize what exactly was analysed. More important, I suggest authors to give rational explanation for performing: "2.7. Anticancer activity (MTT-assay)" and "2.10. Hemolytic activity assay" as I do not see a rationale for these analyses. Although you gave the explanation for "2.10. Hemolytic activity assay" in lines 167-169, I do not agree with given reasons.

- Lines 68-70: Please consider is this the right plase for this sentence. I found the same info in next (M&M) section, under 2.1. „Chemicals“ (where suit perfectly). 

MATERIAL AND METHODS:

- Line 79: „100 bees per location“; From 100 bees you could have collected max. 200 microliters of hemolymph. I am wondering how such a small amount was enough for all the analyses  performed in this study.

- Lines 83-84: „The bees were kept at -20 °C until hemolymph collection“; How long the bees were kept at -20 C and why; Whose methodology did you follow? In case you used new methodology, you should have emphasized that and described it in details.

- Lines 87-89: You refer to Ref. 15., but Cabbri et al. (2018) collected hemolymph by inserting glass microcapillary between fourth and fifth tergite. Please do not refer to Cabbri et al. (2018) since you did not follow the methodology described in that paper.

- Line 90: „The hemolymph samples“; How much hemolymph did you collected per bee? What was the overall volume of hemolymph collected per location? Besides, you lyophilized hemolymph. In that process, you lost over 90% of  weight. However, when describing the analyses, you either did not mention the volume and concentration of the analyzed sample, or you stated only the volume or only the concentration.

if we go in order, it looks like this:

- neither volume nor concentration is specified for method 2.4. in Protein Estimation“;

- neither volume nor concentration is specified for method 2.5. in SDS-polyacrylamide gel electrophoresis“;

- only concentration is given (5 mg/ml), but not exact volume in „2.6. HPLC (High performance liquid chromatography) analysis“;

- only concentrations were given (1000, 500, 250, 125, 62.5, 31.25 μg/ml), but not exact volumes in „2.7. Anticancer activity (MTT-assay)“;

- only volume is given (20-100μL), but not concentration in „2.8. Antimicrobial activity screening“;

- only volume is given (0.3 ml), but not concentration in „2.9. DPPH radical scavenging assay“;

-  neither volume nor concentration is specified (only „series of various concentrations“ is written)  in „2.10. Hemolytic activity assay“.

Therefore, the amount of hemolymph used in the analyses is invisible. Please enter the volume of hemolymph (microL) that you collected from the bees, the amount (mg) that you obtained from the lyophilization process, as well as the amount of used hemolymph in analyzes:

 „2.4. Protein Estimation",

„2.5. SDS-polyacrylamide gel electrophoresis

2.6. HPLC (High performance liquid chromatography) analysis

„2.7. Anticancer activity (MTT-assay)“

2.10. Hemolytic activity assay“

CONCLUSION:

As in the abstract, the main conclusion (last sentence) is prosaic considering the numerous results obtained in this study.

Author Response

Response to Reviewer 1

Abstract Comment: Abstract is weak because it does not reflect what was done and obtained in the study. For example, authors only mentioned „biological activities “without explanation of multiple strong analyses. I also do not agree that the analyses of hemolymph done in this study present „a cheap, simple, yet powerful approach“(they are not cheap and simple). Abstract has to be improved by incorporating exact details of the study. Moreover, I am wondering why the only conclusion is: that hemolymph samples can be used as reliable indicators of bee nutrition and health.

Response: Thank you for this important comment. Following your suggestions we re-wrote the abstract which now reads: “Pollinator declines have raised major concerns for the maintenance of biodiversity and food security, calling for a better understanding of environmental factors that affect their health. Here we used hemolymph analysis to monitor the health status of Western honey bees Apis mellifera. We evaluated the intraspecific proteomic variations and key biological activities of hemolymph of bees collected from four Egyptian localities characterized by different food diversities and abundances. Overall the lowest protein concentrations and the weakest biological activities (cytotoxicity, anti-microbial and anti-oxidant properties) were recorded in hemolymph of bees artificially fed sucrose solution and no pollen. By contrast, the highest protein concentrations and biological activities were recorded in bees that had the opportunity to feed on various natural resources. While future studies should expand comparisons to honey bee populations exposed to more different diets and localities, our results strongly suggest hemolymph samples can be used as reliable indicators of bee nutrition”

Introduction Comment: 1. Line 64: Instead of „some biological activities“, I suggest authors to itemize what exactly was analyzed. More important, I suggest authors to give rational explanation for performing:

Response: We agree and now justify these analyses. Briefly, antimicrobial and antioxidant activities of hemolymph may help assess the health status of honey bees. Anticancer and hemolytic activities may be important for potential therapeutic applications in humans, as suggested by recent observations (Mokarramat-Yazdi et al., 2021).

We changed the title (now “Intraspecific variability in proteomic profiles and biological activities of the honey bee hemolymph”) and the abstract (see our response to your comment above) to better reflect the importance of the bioactivity analyses in our work.

We also significantly changed the introduction. For instance, text L43-213 now reads: “Like all insects, bees have an open circulatory system composed of hemolymph, which is the equivalent of blood in higher vertebrates [5]. Hemolymph is vital to bees as it mediates the distribution of nutrients through the body by supplying tissues and organs [6]. It is the main site for defense against infections [7], and possesses antimicrobial and antioxidant activities [16]. The main components of hemolymph are water, carbohydrates, proteins, inorganic salts, lipids, hormones and immune cells. Proteomic composition, in particular, can inform about the physiological and immune status of bees [5,8]. Inter-individual variations are strongly linked to their nutrition of bees, since food is the major contributor to hemolymph proteins [9–11]. For instance, high hemolymph protein levels minimize the susceptibility of honey bees to pathogens [12,13]. By contrast, low hemolymph protein levels are a signature of poor health status. Therefore, hemolymph analysis can serve as a practical and powerful means for monitoring bee health status. A better understanding of hemolymph variations across individuals and populations may also help better assess its value as potential therapeutic compound as recently suggested (i.e. anticancer activity [66]).”

Text L227-229 at the end of the introduction now reads: “We extracted hemolymph samples from 400 honey bees (100 per location) using two different solvents, and quantified the proteomic content and potential anticancer, antibacterial, antioxidant, and hemolytic activities.”

Comment: "2.7. Anticancer activity (MTT-assay)" and "2.10. Hemolytic activity assay" as I do not see a rationale for these analyses. Although you gave the explanation for "2.10. Hemolytic activity assay" in lines 167-169, I do not agree with given reasons.

Response: Please see our response to your comment 1.

Introduction Comment: 2. Lines 68-70: Please consider is this the right place for this sentence. I found the same info in next (M&M) section, under 2.1. „Chemicals“ (where suit perfectly).

Response: The sentence was removed from the introduction.  

Materials and methods Comment: 3. Line 79: „100 bees per location“; From 100 bees you could have collected max. 200 microliters of hemolymph. I am wondering how such a small amount was enough for all the analyses performed in this study.

Response: The hemolymph was collected by puncturing the bee body according to (Łoś and Strachecka, 2018) and (Basseri et al., 2016). Using this approach, it is estimated that ca. 6-20 µl can be extracted per bee (Łoś and Strachecka 2018;  Migdał et al. 2020). In our experiments, we extracted 9-14 µl of hemolymph per bee, i.e. 1000-1200 µl per site after lyophilization.

This was clarified on L486-493: “Hemolymph was extracted according to [18,19]. Here, however, we sampled near the coxal membrane using sterile insulin syringes and pressing the abdomen of bees. Clear and slightly yellow hemolymph was drawn out from the wound. If cloudy yellow intestinal contents were taken, the sample was discarded. The hemolymph samples were kept into sterile Eppendorf tubes and stored at -20 °C until lyophilization. About 9-14 µl of hemolymph was extracted per bee. The hemolymph of all the bees from a given site was then pooled, yielding about 1000-1200 µl per location. After lyophilization, this resulted in about 100 mg per location. Equal weights of lyophilized hemolymph were dissolved in DMSO or PBS.”

Materials and methods Comment: 4. Lines 83-84: „The bees were kept at -20 °C until hemolymph collection“; How long the bees were kept at -20 C and why; Whose methodology did you follow? In case you used new methodology, you should have emphasized that and described it in details.

Response: Text L248-249 now reads: “The bees were then cold anaesthetized (kept at -20 °C) for 5 min to facilitate hemolymph extraction on live individuals [17]”.

Materials and methods Comment: 5. Lines 87-89: You refer to Ref. 15., but Cabbri et al. (2018) collected hemolymph by inserting glass microcapillary between fourth and fifth tergite. Please do not refer to Cabbri et al. (2018) since you did not follow the methodology described in that paper.

Response: We removed this reference from the methods.

Materials and methods Comment: 6. Line 90: „The hemolymph samples“; How much hemolymph did you collected per bee? What was the overall volume of hemolymph collected per location? Besides, you lyophilized hemolymph. In that process, you lost over 90% of weight. However, when describing the analyses, you either did not mention the volume and concentration of the analyzed sample, or you stated only the volume or only the concentration. If we go in order, it looks like this:

Response:  Please see our response to your comment 3 regarding quantities of hemolymph extracted.

Comment: - neither volume nor concentration is specified for method 2.4. in „Protein Estimation“;  

Response: Text L496-499 now reads “Protein concentration was measured in mg/ml using a Thermo Scientific™ Nano Drop™ One Micro volume UV-Vis Spectrophotometer. Bovine serum albumin (BSA) was added as a standard in each sample (2µl, 1mg/ml) [20]. Three samples were analysed for a given site.”

Comment: - neither volume nor concentration is specified for method 2.5. in “SDS-polyacrylamide gel electrophoresis“;

Response: Text L508-510 now reads: “The samples were immediately loaded into wells (15 µL of sample with a protein concentration of 100 µg/ml was added in each well).”

Comment: - only concentration is given (5 mg/ml), but not exact volume in „2.6. HPLC (High performance liquid chromatography) analysis“;

Response: Text L516-523 now reads: “Reversed Phase HPLC chromatograms of the honey bee hemolymph extracts separate the protein peaks based on their retention time. Each of the lyophilized hemolymph samples was dissolved in either solvent (PBS or DMSO) at the same concentration (5 mg/ml). 70 µl samples from each location were analysed using YL9100 HPLC System under the following conditions: C18 column (Promosil C18 Column 5 μm, 150 mm×4.6mm) as stationary phase and acetonitrile (ACN) gradients in the range of 10% to 100% acetonitrile in water for 50 min as mobile phase at flow rate 1 ml/min and by using UV detector at wavelength 280 nm [19].”

Comment: - only concentrations were given (1000, 500, 250, 125, 62.5, 31.25 μg/ml), but not exact volumes in „2.7. Anticancer activity (MTT-assay)“;

Response: Text 651-653 now reads : “After incubation, the cells were treated with serial concentrations of 100 μlhemolymph extracts in each solvent (1000, 500, 250, 125, 62.5, 31.25 µg/ml) and then incubated at 37 °C in 5% CO2atmosphere for 48 h”

Comment: - only volume is given (20-100μL), but not concentration in „2.8. Antimicrobial activity screening“;

Response: Text L674-676 now reads : “A volume of about 20-100µL of each extract (10 mg/ml) or control at known concentration was introduced into the well.”

Comment: - only volume is given (0.3 ml), but not concentration in „2.9. DPPH radical scavenging assay“;

Response: Text LX now reads: “Briefly, 100μl of DPPH methanolic solution (0.004% in methanol) were mixed with 0.3 ml of sample extracts (1 mg/ml) or standard and incubated in dark for 30-60 min at 25 °C.”

Comment: - neither volume nor concentration is specified (only „series of various concentrations“ is written)  in „2.10. Hemolytic activity assay“.

Response: Text L685-687 now reads “We assessed the ability of the hemolymph extracts (100 μL) from concentrations (125, 250, 500, 1000, 2000, 4000 µg/ml) to lyse human erythrocytes.”

Therefore, the amount of hemolymph used in the analyses is invisible. Please enter the volume of hemolymph (microL) that you collected from the bees, the amount (mg) that you obtained from the lyophilization process, as well as the amount of used hemolymph in analyzes:

 „2.4. Protein Estimation",

„2.5. SDS-polyacrylamide gel electrophoresis“

„2.6. HPLC (High performance liquid chromatography) analysis“

„2.7. Anticancer activity (MTT-assay)“

„2.10. Hemolytic activity assay“?

Response: See our responses above.

Conclusion Comment: 7. As in the abstract, the main conclusion (last sentence) is prosaic considering the numerous results obtained in this study.

Response: Conclusion L1544-1555 now reads: “Our comparative analysis reveals significant intraspecific variability in the composition and concentration of hemolymph of honey bees sampled in Egypt. The hemolymph of the honey bee that could feed on cultivated plants and wild medicinal plants possessed relatively higher protein concentrations, as well as higher anti-microbial, antioxidant and anti-cancer properties compared to that of bees reared only on sugar solutions in artificial conditions. Future studies should expand our approach to more localities and diets, in order clarify the link between diet and hemolymph properties, as well as a potential therapeuthic usage of honey bee hemolymph.”

Reviewer 2 Report

Honeybee nutrition and its relationship with health is an interesting topicof growing importance. The authors approach it by measuring total protein and comparing SDS-PAGE bands and HPLC spectra of hemolymph of bees from three free-flying origins of bees, and a group hel in tents (no access to pollen).

The overall concept of the study is very interesting, but there are a number of shortcomings both with regard to the parameters observed and with regard to data analysis. These should be remedied before the manuscript is published:

-        The statistical analysis (multiple t-tests for comparison of the 4 groups instead ogf ANOVA + post-hoc test) is inappropriate

-        It is unclear why anti-cancer properties as well as haemolytic properties of HL are measured. In my opinion, these measurements should eiher be left out, or announced/explained in the introduction.

-        A number of important details are lacking in the description of the methodology:

o   Were bees from the different locations of the same age, subspecies, genetics, were they sampled during the same season?

o   For how long were colonies on site A deprived of pollen?

o   Was the HL of the 100 bees of each site pooled before analysis, or were analyses repeated 100 times? What is the number of cases in the analyses, and what does each case represent?

L17: higher

L32: as you only studied influences of nutrition on HL composition, you should not speak of health here

L37-39: please restructure this sentence, there seems to be a verb lacking

L9: indicators of

L52: are linked

L56: has shown

L57: are you certain you do not mean “between castes”?

L83: Freezing before hemolymph extraction could potentially lead to a protein denaturation or mixture of hemolymph with liquids from other body compartnments and tissues – is there any literature showing that freezing before HL extraction is without influence on results of proteomic analyses?

L117: as cancer does not play any significant role in short-living animals like honeybee foragers, I do not think this analysis is of interest here

L166-168: The justification given for the hemotolysis assay ( to check whether extacts are safe for medical use) seems unrelated with the research question as pointed out in the introduction – so either modify the research question, or remove the section on this assay

L182: statistical analysis: as more than two groups were compared, the t-test is not approproíate here – please use ANOVA

Methods: please indicate wheher bees from the different locations are genetically similar (same subspecies?) or not

L350-376: given that you only measured protein weight, and only at a relatively low resolution, identification of bands is highly speculative. It’s ok to hypothesize on identity of certain bands, but you should make it clear that these are only speculations.

L391: again, anti-cancer and haemolytic properties of hemolymph are not part of the research question the authors defined in the introduction

Author Response

Response to Reviewer 2

General Comment: Honeybee nutrition and its relationship with health is an interesting topic of growing importance. The authors approach it by measuring total protein and comparing SDS-PAGE bands and HPLC spectra of hemolymph of bees from three free-flying origins of bees, and a group held in tents (no access to pollen).

The overall concept of the study is very interesting, but there are a number of shortcomings both with regard to the parameters observed and with regard to data analysis. These should be remedied before the manuscript is published:

Response: Thank you for the positive comment.

Comment: 1.The statistical analysis (multiple t-tests for comparison of the 4 groups instead of ANOVA + post-hoc test) is inappropriate.

Response: We apologies for the confusion. Our approach is now clarified at L805-811: “All statistical analyses were done in SPSS 22.0. Pairwise comparisons between parameters of hemolymphs extracted in PBS and DMSO were made with Student’s t-tests. Multiple comparisons between parameters of hemolymphs from bees of the four study sites were made with One-way ANOVAs followed by a Tukey’s test. When the P-value was lower than 0.05, the difference in the means of the samples were considered as statistically significant. Means are reported with standard errors (mean ± SE).”

Comment: 2.It is unclear why anti-cancer properties as well as haemolytic properties of HL are measured. In my opinion, these measurements should either be left out, or announced/explained in the introduction.

Response: We agree and now justify these analyses. Briefly, antimicrobial and antioxidant activities of hemolymph may help assess the health status of honey bees. Anticancer and hemolytic activities may be important for potential therapeutic applications in humans, as suggested by recent observations (Mokarramat-Yazdi et al., 2021).

We changed the title (now “Intraspecific variability in proteomic profiles and biological activities of the honey bee hemolymph”) and the abstract (see our response to your comment above) to better reflect the importance of the bioactivity analyses in our work.

We also significantly changed the introduction. For instance, text L43-213 now reads: “Like all insects, bees have an open circulatory system composed of hemolymph, which is the equivalent of blood in higher vertebrates [5]. Hemolymph is vital to bees as it mediates the distribution of nutrients through the body by supplying tissues and organs [6]. It is the main site for defense against infections [7], and possesses antimicrobial and antioxidant activities [16]. The main components of hemolymph are water, carbohydrates, proteins, inorganic salts, lipids, hormones and immune cells. Proteomic composition, in particular, can inform about the physiological and immune status of bees [5,8]. Inter-individual variations are strongly linked to their nutrition of bees, since food is the major contributor to hemolymph proteins [9–11]. For instance, high hemolymph protein levels minimize the susceptibility of honey bees to pathogens [12,13]. By contrast, low hemolymph protein levels are a signature of poor health status. Therefore, hemolymph analysis can serve as a practical and powerful means for monitoring bee health status. A better understanding of hemolymph variations across individuals and populations may also help better assess its value as potential therapeutic compound as recently suggested (i.e. anticancer activity [66]).”

Text L227-229 at the end of the introduction now reads: “We extracted hemolymph samples from 400 honey bees (100 per location) using two different solvents, and quantified the proteomic content and potential anticancer, antibacterial, antioxidant, and hemolytic activities.”

Methodology Comment: 3.Were bees from the different locations of the same age, subspecies, genetics, were they sampled during the same season?

Response: This section has been expanded and now reads L239-249 “We exclusively worked on forager honey bees (hybrids of the subspecies A. mellifera lamarckii, A. mellifera carnica and A. mellifera ligustica). Sweep nets were used to collect honey bee foragers of unknown age in the different study sites (ca. 100 bees per location) from May to July 2021. Foragers were caught while flying around food sources. Bees from each site were collected on the same day. Bees from the faculty of science of Port Said Governorate originated from colonies enclosed in a big flight tent and maintained in an artificial beehive with access to sugar syrup only for the duration of the experiment (from beginning of May to end of June). Bees from Ismailia, Suez governorates and Saint Catherine had access to different cultivated plants (see details in Table 1), thus providing diverse nutritional resources. The bees were then cold anaesthetized (kept at -20 °C) for 5 minutes to facilitate hemolymph extraction on live individuals [17].”

Methodology Comment: 4.For how long was colonies on site A deprived of pollen? 

Response: See our response to your comment 3.

Methodology Comment: 5.Was the HL of the 100 bees of each site pooled before analysis, or were analyses repeated 100 times? What is the number of cases in the analyses, and what does each case represent?

Response: Yes hemolymph of the 100 bees of each site was pooled before the analysis. Text L490-493 now reads: “About 9-14 µl of hemolymph was extracted per bee. The hemolymph of all bees from a given site was then pooled, yielding about 1000-1200 µl per location. After lyophilization, this resulted in about 100 mg per location.”

Methodology Comment: 6.

L17: higher 

Response: Done L20.

L32: as you only studied influences of nutrition on HL composition, you should not speak of health here

Response: We removed mention to health.

L37-39: please restructure this sentence, there seems to be a verb lacking

Response: Sentence L40-43 now reads “Over the past decades, their accelerated losses have called for a better understanding of the environmental stressors that affect bee population growth and their mechanisms of action, for instance through large-scale monitoring of colony health statuses across different habitats [4].”

L9: indicators of  

Response: Sentence L204-206 now reads “Proteomic composition, in particular, can inform about the physiological and immune status of bees [5,8].”

L52: are linked

Response: Sentence L206-207 now reads “Inter-individual variations are strongly linked to the nutrition of bees, since food is the major contributor to hemolymph proteins [9–11].”

L56: has shown

Response: Sentence L205-207 now reads “Previous research has shown that the hemolymph of the honey bee Apis mellifera varies in protein composition across individuals of the same population, especially within castes and among different developmental stages [14].”.

Methodology Comment: 7.L57: are you certain you do not mean “between castes”?

Response: Yes, see our response to your comment 3.

Methodology Comment: 8.L83: Freezing before hemolymph extraction could potentially lead to a protein denaturation or mixture of hemolymph with liquids from other body compartnments and tissues – is there any literature showing that freezing before HL extraction is without influence on results of proteomic analyses?

Response: We cold anaesthetized bees for 5 min (see our response to you comment 3). We believe this procedure causes minimal damages to hemolymph.

Methodology Comment: 9.L117: as cancer does not play any significant role in short-living animals like honeybee foragers, I do not think this analysis is of interest here

Response: See our response to your comment 2.

Methodology Comment: 10.L166-168: The justification given for the hemotolysis assay ( to check whether extracts are safe for medical use) seems unrelated with the research question as pointed out in the introduction – so either modify the research question, or remove the section on this assay

Response: See our response to your comment 2.

Methodology Comment: 11.L182: statistical analysis: as more than two groups were compared, the t-test is not appropriate here – please use ANOVA

Response: See our response to your comment 1.

Methods Comment: 12.Please indicates whether bees from the different locations are genetically similar (same subspecies?) or not

Response: See our response to your comment 3.

Methods Comment: 13.L350-376: given that you only measured protein weight, and only at a relatively low resolution, identification of bands is highly speculative. It’s ok to hypothesize on identity of certain bands, but you should make it clear that these are only speculations.

Response: Thank you. In the discussion the protein bands were mentioned as being “in the same range” or “evoke” or “are likely”, not as if they were for sure same proteins.

Methods Comment: 14.L391: again, anti-cancer and haemolytic properties of hemolymph are not part of the research question the authors defined in the introduction

Response: See our response to your comment 2.

Round 2

Reviewer 1 Report

The manuscript is greatly improved. I am satisfied with authors' responses. There are only some minor technical mistakes: 

1) reference numbering: The reference below is duplicated in reference list (15) and (66), so it is necessary to clarify its number and check the reference to it in the text.

- Mokarramat-Yazdi et al. J. Adv. Pharm. Educ. Res. 2021, 10, 72–78.

2) inadequate application of brackets in several places, e.g.: 

-  lines 201-202: "from concentrations (125, 250, 500, 1000, 2000, 201 4000 μg/ml)"

- line 108: Hemolymph was extracted according to [19,20].

Author Response

Thank you for the positive assessment of our revised manuscript. We explain below how we have dealt with each of the remaining minor comments. We also carefully proof reread the text for English with the help of a native English speaker in the lab.

1) reference numbering: The reference below is duplicated in reference list (15) and (66), so it is necessary to clarify its number and check the reference to it in the text.

- Mokarramat-Yazdi et al. J. Adv. Pharm. Educ. Res. 2021, 10, 72–78.

Our response: Thank you. Mokarramat-Yazdi et al. 2021 is now exclusively refereed as (15).

2) inadequate application of brackets in several places, e.g.:

- lines 201-202: "from concentrations (125, 250, 500, 1000, 2000, 201 4000 μg/ml)"

Our response: Done.

- line 108: Hemolymph was extracted according to [19,20].

Our response: Done.